# Parents’ or Guardians’ Decisions on Human Papillomavirus Vaccine Acceptance for School Children in a Southern Province of Thailand: A Mixed-Method Study

**DOI:** 10.3390/vaccines14010053

**Published:** 2025-12-31

**Authors:** Thanyalak Thongkamdee, Supinya Sono, Chutarat Sathirapanya

**Affiliations:** 1Department of Family and Preventive Medicine, Faculty of Medicine, Prince of Songkla University, Hat Yai, Songkhla 90110, Thailand; tthanyal@medicine.psu.ac.th (T.T.); supinya.so@psu.ac.th (S.S.); 2Research Center for Kids and Youth Development, Prince of Songkla University, Hat Yai, Songkhla 90110, Thailand

**Keywords:** cervical cancer, vaccine, human papillomavirus, knowledge, acceptance

## Abstract

**Background:** Cervical cancer is associated with Human Papillomavirus (HPV) infection. Besides cervical cancer, oro-pharyngo-laryngeal or uro-genital cancers are also reported. The HPV vaccine has been strongly recommended for school age children. However, the parents’ or guardians’ hesitancy remains. **Methods:** This is a mixed-method study in which the parents or guardians of school children, aged 10–18 years, were enrolled voluntarily. Their general demographic data, knowledge, attitudes, and awareness of vaccine accessibility, healthcare cost entitlement of the children, types of school affiliation, education administration areas where the schools were located, and the presence of a healthcare professional in family were analyzed by multiple logistic regression analysis adjusted with all studied variables to define the significant associated factors with the parents’ or guardians’ HPV vaccine acceptance (*p* < 0.05). In-depth interviews were subsequently performed with the selected participants until the qualitative data were saturated. Thematic analysis was applied, and the results of the two study methods were integrated to explore the reasons for vaccine acceptance or hesitancy. **Results:** A total of 943 questionnaire respondents were enrolled, among whom 75.8% were female and 86.4% were parents. A total of 663 (70.3%) participants accepted the HPV vaccine. Parents’ or guardians’ knowledge and attitudes, awareness of vaccine accessibility, type of school affiliation, the children’s healthcare cost entitlement, and the presence of a healthcare professional in the family were significantly associated with vaccine acceptance in the multivariate analysis (*p* < 0.05). The qualitative study revealed that misunderstanding of the vaccine’s safety and benefits combined with inadequate reliable information sources were associated factors with HPV vaccine hesitancy among the parents or guardians. **Conclusions:** Providing clear-cut knowledge about the HPV vaccine benefit vs. risk and clearing financial barriers for the parents or guardians of school children are advocated.

## 1. Introduction

Cervical cancer is the fourth most common cancer in women globally with around 660,000 new cases and around 350,000 deaths in 2022 [1]. In Thailand, cervical cancer has been the third most common cancer among women. Current estimates indicate that every year 9158 women are diagnosed with cervical cancer and 4705 die from the disease. About 3.4% of women in the general population are estimated to acquire Human Papillomavirus (HPV) 16 or 18 infections at a given time [2].

HPV infection is a kind of sexually transmitted disease. Although it commonly causes cervical cancers in females, oral cavity, pharyngo-laryngeal, or uro-genital cancers related to the viral infection can be found in both genders [2]. Particularly, HPV types 16 and 18 have been recognized as the pathogens highly associated with cervical cancer, accounting for about 70% [3,4,5]. The infection precedes and remains sub-clinically for many years before the development of cervical or other related cancers. The efficient prevention of contracting the HPV infection among virgin pre-adolescent children and intensive cervical cancer screening among sexually active adults can limit the annual incidence and death rate of cervical cancer in particular.

The HPV vaccine was approved by the US-FDA for the prevention of HPV infection in 2006. Thence, it was included in the national vaccine programs of 146 countries [6]. In 2014, WHO guidelines on HPV vaccination programs recommended a two-dose regimen of the HPV vaccine with 6 months apart for girls aged 9 to 14 years due to their comparable efficacy with the three-dosage regimen [7,8]. Despite the wide availability of the HPV vaccine, knowledge about the HPV vaccine, vaccine cost (self-payment or governmental funding), anti-vaccine fake news, religious and cultural beliefs, concerns about infertility and promiscuity, and routes of vaccine access contributed to HPV vaccine hesitancy [9,10,11,12,13,14,15,16,17]. Additionally, the HPV vaccine prescribed to male children for the prevention of HPV-related cancers other than cervical cancer remains very doubtful for most parents [18].

In Thailand, bivalent and quadrivalent HPV vaccines were listed in the national vaccination program in 2017. Two doses of the vaccine with a 6-month interval were recommended for females aged 11–12 years [6]. The target of two doses of vaccine coverage was aimed at >90% by the Ministry of Public Health of Thailand. However, a study showed an overall vaccine coverage of 60% in 2018, and a decremental trend through 2022 due to the common causes of vaccine hesitancy found in the literature and the shortage of vaccine supply in 2019 and COVID-19 pandemic in 2021 [19].

In this mixed-method study, we explored the general demographics, knowledge, and attitudes regarding HPV infection and vaccine, vaccine accessibility, and the socio-economic status, educational level, religious beliefs, and social norms of the parents or guardians of primary (grades 1–6) and secondary (grades 7–12) school children aged 10–18 years in a southern province of Thailand. The associations of the parents’ or guardians’ characteristics, knowledge, attitudes, and barriers of access to the vaccine with their decision to accept HPV vaccination for their children were analyzed. The results of both study methods were aimed to be integrated and applicable as guidelines to design and implement health education programs to facilitate HPV vaccine acceptance among the parents or guardians.

## 2. Materials and Methods

### 2.1. Study Design and Setting

This is a sequential and mixed-method study designed to explore the decision of the parents or guardian to accept the HPV vaccine for their children aged 10–18 years who were studying in the selected schools of Songkhla province. We assessed the associations of the parents’ or guardians’ characteristics and vaccine-related factors with their decisions to accept HPV vaccination for their children in a quantitative study.

Semi-structured interview questions were used to collect the data of the qualitative study by a subsequent in-depth interview. The collected data underwent thematic analysis to identify the major themes and provide a systematic interpretation of the implications.

### 2.2. Study Participants

We enrolled parents or guardians aged ≥25 years old who provided care to the school children, aged 10–18 years old, who were studying and living in Songkhla province. They were able to understand both spoken and written Thai well.

### 2.3. Sample Size Calculation

To evaluate HPV vaccine acceptance by the parents or guardians of school children aged 10–18 years, the formula for estimating population proportion was used.*n* = [*z*^2^_1−α/2_
*p*(1 − *p*)]/*d*^2^

[Proportion (*p*) = 0.5, Alpha (α) = 0.05 (confidence level), Error (*d*) = 0.05 (acceptable margin of error)].

Given that the largest sample size required was 385 and stratified and cluster sampling methods were used (design effect = 2), the adjusted sample size was 770. Finally, the required sample size was 847 participants after adding 10% more for missing data.

### 2.4. Sampling Techniques

In the quantitative study, stratified and clustered sampling techniques were used. We started by contacting the Songkhla Provincial Education Authority to ask for permission to conduct the study. A school from each authorized administration system, or school affiliation, i.e., central governmental school (CGS), local governmental school (LGS), or private school (PS), in four provincial education administration areas (areas 1–4) was sequentially randomized and invited to participate the study. One school per each authorized administration system was required to represent the group. There were three schools per each education administration area that would be enrolled upon invitation acceptance. Once a total of 12 schools agreed to participate, the researcher team approached the parents or guardians in each school through regular school meetings to invite them to participate in this study on their voluntariness. When the written consents were signed by the invitation-accepted parents or guardians, the data collection process began.

For the qualitative study, we purposively enrolled the parents or guardians who participated in the quantitative study for the further in-depth interview. Around 10–30 participants were enrolled depending on interview data saturation. Written informed consent for participating in the interview and audio recording were obtained as well.

### 2.5. Study Instruments

#### Questionnaire Development and Validation

In the quantitative study, the case record form was structured into six sections: (a) demographic information, (b) knowledge regarding HPV and its vaccine, (c) vaccine accessibility, (d) attitudes and concerns about HPV vaccination for the children, (e) decision on HPV vaccination for the children, (f) prior vaccination experiences. The questionnaires for the evaluation of (b) through (d) were self-developed by the researcher team based on the key constructs of the Health Belief Model (HBM), which encompasses perceived susceptibility, perceived severity, perceived benefits, and perceived barriers, and cue to action. The content validity of the developed questionnaires was assessed by a panel of three experts in obstetrics and gynecology, family medicine, and preventive medicine. The item-objective congruence (IOC) scores were 0.909 for (b), 1.0 for (c), and 0.902 for (d), and the overall IOC score was 0.933. The content reliability of the questionnaires tested among 30 parents residing in Songkhla Province showed 0.701 for (b), 0.802 for (c), and 0.709 for (d) by Cronbach’s alpha coefficients. The scoring methods and cut-points for each questionnaire were stated in the footnote of the questionnaires (See Appendix A). We classified level of knowledge and attitudes based on Bloom’s cut-off classification [20], while that of accessibility was based on Best’s cut-off classification [21].

The qualitative interviews were conducted through the semi-structured interview format by using the core questions derived from the participants’ perspectives and responses during data derivation in the quantitative study. The author (T.T.) conducted the interview process. The recorded interview content was subsequently transcribed for thematic analysis.

### 2.6. Data Collection

During the school meetings between parents or guardians and school teachers, the researcher team explained the study details and process to the parents or guardians of 11- to 18-year-old children (who were studying in grades 5–12) to ask for their consent. After the written informed consents were obtained, we enrolled the study participants and distributed printed or online-based questionnaires to them. We collected the general characteristics, the type of relationship with the children (parent vs. non-parent), types of school in which the children studied (CGS, LGS, or PS), healthcare cost entitlement, and parents’ or guardians’ experience of receiving a vaccine. All responders’ personal information was fully anonymized. The data obtained were securely stored and access only by the researcher team was permitted.

After completion of the quantitative data analysis, the researchers developed and outlined the questions for the in-depth interview of the qualitative study. We purposively selected participants with a variety of baseline characteristics who completed the questionnaires of the quantitative study. Finally, 10 to 30 voluntary participants were enrolled depending on the number of data required for saturation. Face-to-face interviews were conducted in private settings and the audio recorded content was transcribed and securely stored with limited access.

### 2.7. Data Analysis

Descriptive data were presented in frequencies and percentages, means (SD), and median (IQR) where appropriate. Factors associated with HPV vaccination decisions were examined using multiple logistic regression. The variables with a *p*-value < 0.20 in univariate analysis were proceeded to the multivariate model. Model selection was performed using the backward stepwise elimination method with statistical significance (*p* < 0.05).

Thematic analysis was applied for the qualitative study to detail the contextual understanding of the quantitative results. To enhance the credibility of the results, multiple analyst triangulation by three independent researchers was applied in the analytical process. The results of both study methods were integrated and discussed for implication. GRAMMS checklist for mixed-method study was applied to ensure the study’s integrity (see Appendix A).

### 2.8. Ethical Approval Statement

The study protocol was reviewed and approved by the institutional ethical committee (EC code No. REC 67-370-9-4, approval date 25 September 2024). We confirmed that the study process strictly complied to the regulations stated in the updated version of the Declaration of Helsinki and its related amendments. All the identifiable personal data of the study participants were completely anonymous. Written informed consents to participate in both phases of the study were obtained.

### 2.9. Data Availability Statement

All data and analysis methods are reported in this manuscript. No data or parts of them were deposited in any pre-publication depository sources.

## 3. Results

### 3.1. General Demographics

There was a total of 943 respondents, of whom 792 (response rate 39.11%) participants responded via printed questionnaires, while 152 (response rate 1.16%) participants responded via online.

Among them, 75.8% were female, 45% aged 40–49 years, and 86.4% were the children’s parents. Most of them (82.9%) believed in Buddhism, while 16.1% were Muslims. The larger group of them earned their living by owning small business enterprises or as private sector employees. A total of 50.8% graduated at Bachelor level and above. The largest group of the parents or guardians had their children studying in CGS (46.2%) and resided in the education administration area 3 of Songkhla Province (Leelawadee Zone: Sadao District, Hat Yai District, Na Mom District, and Khlong Hoi Khong District) (55.5%).

The data revealed that 663 participants (70.3%) accepted the HPV vaccination for their children, but only 277 children (29.3%) had already been vaccinated. Girls were vaccinated more than boys (16 cases, 1.7%).

The significant demographic factors associated with parents’ or guardians’ acceptance of HPV vaccination for their children included the education administration area of the school (*p* < 0.001), the type of school attended (*p* < 0.001), the education level of parents or guardians (*p* = 0.018), the healthcare cost entitlement of the children (*p* < 0.001), the presence of a healthcare professional in the family (*p* = 0.002), and the parent’s own experience with a vaccination (*p* < 0.001) (Table 1).

### 3.2. Knowledge About HPV and HPV Vaccine

The questionnaire evaluating knowledge of HPV and the HPV vaccine comprised 14 true-or-false questions. There was a significant difference in the median knowledge scores between the parents who accepted the vaccine and those who did not (10 (9,11) vs. 10 (8,10); (*p* < 0.001)).

When looking at the association of an individual item of knowledge with parents’ acceptance, it was found that seven items had statistically significant associations. These included the following: understanding that HPV stands for Human Papillomavirus (*p* = 0.008), recognizing that HPV can cause cancers of the oral cavity, vagina, anus, and other areas (*p* = 0.003), knowing that HPV infection takes more than 10 years to develop to be a cancer (*p* < 0.001), and understanding the safety and efficacy of the HPV vaccine in preventing HPV-related cancers (*p* = 0.003), children aged 9–14 years require only two doses of vaccines (*p* < 0.001), vaccine is most effective when administered before the first sexual relation (*p* < 0.001), and HPV vaccine prevents only cervical cancer (*p* = 0.033) (see Appendix A).

### 3.3. Access to Information and Service of HPV Vaccine

Knowing about the presence of the HPV vaccine (*p* < 0.001), receiving a recommendation from a healthcare professional (*p* < 0.001), and awareness of the availability of the government-supported HPV vaccination program (*p* < 0.001) and where to receive a HPV vaccine near the respondent’s residence area (*p* < 0.001) were associated with parents’ or guardians’ acceptance of HPV vaccination for their children. In contrast, perceived barriers such as vaccine cost, travel distance, or lack of information had no statistically significant association with the vaccine acceptance (*p* = 0.107) (item 5 of the questionnaire). The median of the composite score of knowing about and how to access the HPV vaccine showed significant association with the acceptance of HPV vaccination for the children (3 (IQR 1,4) vs. 1.5 (IQR 1,3), (*p* < 0.001)) (see Appendix A).

### 3.4. Attitudes and Concerns About HPV Vaccine

Most parents believed that HPV vaccine is essential for cancer prevention (84.2%). However, the current public campaigns were insufficient (70.2%) for emphasis. In addition, 72.0% of parents agreed that males should also be encouraged to receive the HPV vaccine and free HPV vaccination should be provided for both genders (85.5%). Despite the positive attitudes, a portion of the parents expressed concerns about the potential side effects of the vaccine (72.4%), and the vaccine cost if free vaccine accessibility was impossible (55.5%). The parents’ acceptance of HPV vaccination for their children was significantly associated with the parents’ attitudes and concerns (*p* < 0.001) (see Appendix A).

### 3.5. Factors Associated with Parental Acceptance of HPV Vaccine for Their Children

The multivariate logistic regression analysis revealed that all variables, except for the education administration area, were significantly associated with parents’ adoption of the HPV vaccine (*p* = 0.05) (Table 2).

### 3.6. Qualitative Study

In the qualitative study, semi-structured interviews were applied for data collection. A total of 11 participants (10 female and 1 male) were interviewed. Two participants had their children studying in schools of the education administration area 1, three were of area 2, and six were of area 3. Most of the participants were the parents of the school children, except only one, who was an aunt. Two of the participants were healthcare professionals as well (see Appendix A). The content of the interview was analyzed by thematic analysis, and its details and quotations for examples are shown (Table 3) (For full quotations, please see Appendix A).

## 4. Discussion

This study found that parents’ knowledge, attitudes, and awareness of the accessibility of the vaccine, the types of affiliation of the schools, the presence of healthcare personnel in family, and healthcare cost entitlement were significantly associated with the parents’ or guardians’ HPV vaccine acceptance for their children in multivariate analysis. Large 95% CIs of attitudes to the HPV vaccine could lessen its significance (Table 2).

The public adoption of newly introduced vaccines commonly faces barriers or oppositions, like that of the COVID-19 vaccine during the global pandemic previously. Safety concerns usually emerge first, particularly when the vaccine is recommended for young children as in the case of the HPV vaccine. Correct knowledge regarding the safety and benefit of the vaccine is strongly required by the parents before making a decision. In our study, it was noted that some study participants did not realize that the HPV vaccine was prescribed for cervical cancer prevention (item 7), while some knew its benefit in the prevention only of cervical cancer but did not include other HPV-induced cancers (item 8). Additionally, we found that “HPV infection cannot cause cancer in males. It is the disease of only females (item 2)” was a common misunderstanding found in both study methods because the common naming of the HPV vaccine in Thailand is a vaccine for cervical cancer prevention, implying that the vaccine should be given to only females. Another knowledge gap of the parents causing HPV vaccine hesitancy was that cervical cancer was not common among the younger-aged female children (see Appendix A). For the attitudes towards the HPV vaccine, there were significant differences in every item tested between the parents who accepted and those did not accept the vaccination. We found that the level of HPV vaccine knowledge affected the attitudes abundantly (see Appendix A). Although a portion of the parents knew about the clinical benefits of the HPV vaccine, discussion and confirmation confirming its safety and benefits with healthcare professionals rather than by printed media or online sources were strongly required. Vaccine safety was a more serious concern than its benefits among the study participants. This issue of strong concern was reported from both study methods needing an explanation for understanding before the acceptance of child vaccination. Confidence about vaccine safety and the benefits were significantly associated with high vaccine acceptance in Thailand [11].

Additionally, payment for the vaccine cost was a significant factor affecting the decision of vaccine acceptance as well. For the children whose healthcare cost was paid by governmental reimbursement (i.e., one or both of their parents was or were governmental officers/employees), their parents did not need to pay vaccine cost at all, while the other entitlement schemes would have a copay. This finding was supported by a systemic review performed in ASEAN countries where vaccine acceptance was higher if it was offered without cost [10].

The similar findings of the qualitative and the quantitative studies were the understanding that the vaccine prevented only cervical cancer and had no benefit in men at all, and that it should be given only when becoming an adult female. Otherwise, data of the qualitative study also highlighted the impact of the vaccine information source like that found in the quantitative study. The parents reported that they relied mostly on the information provided by healthcare professionals through face-to-face discussions or explanations. A study that evaluated the factors associated with the parents’ decision of vaccine acceptance for their 9–12-year-old children in China reported that healthcare professionals or official healthcare agencies were the most reliable sources of vaccine information [18]. A pediatrician, if available, was requested by one of the interviewed participants to provide the vaccine information. Alternatively, tailored-trained nurses or community healthcare personnel on HPV vaccination were suggested to take this role [22]. We believed that the available school nurses or community healthcare workers distributed over Thailand were competent enough to respond to this task.

Integrated consideration of the qualitative with the quantitative data provided clearer insights about the rationales of the understanding and practices of the study parents, and why HPV vaccine hesitancy remained. What the results of the qualitative study added to the quantitative study included (a) the misnomer of the HPV vaccine caused the misunderstanding that the vaccine prevented only cervical cancer but not the others related to HPV infection, (b) the vaccine, therefore, was useful for only females, (c) children of primary to secondary school age were too young to receive the vaccine due to the parents’ concerns of the vaccine’s safety, (d) no adequate reliable sources of knowledge to ensure vaccine efficacy and safety, and the currently available vaccine information was mostly from non-official public media or printed materials that were unable to give more required details, and therefore healthcare professionals are strongly needed as the vaccine information providers, (e) reimbursement or financial support of vaccine cost for people who could not afford it was limited, and (f) the method of vaccine access, being community- or school-based, should be clearly specified by the national healthcare policy. These points altogether were associated with vaccine hesitancy and necessitated collaborative and systematic modifications.

## 5. Strengths and Limitations

The current study applied a mixed-method study to explore the association of knowledge, attitudes towards HPV vaccine, vaccine accessibility, and the parents’ or guardians’ characteristics with the decision of HPV vaccine acceptance for their children living in a metropolitan city of southern Thailand. The qualitative method further added clearer insights into the parents’ understanding about the HPV vaccine and barriers of access. We intended to demonstrate the facilitators and barriers of early HPV vaccination in young children in alignment with the standard guideline for HPV-related cancer prevention. Nevertheless, the results were drawn from a limited study group and area, which will limit their generalizability.

## 6. Conclusions

Many misunderstandings caused by a lack of clear-cut vaccine knowledge and reliable information sources were significantly associated with vaccine hesitancy in this study. Co-ordination between education and healthcare sectors to provide the correct knowledge of HPV-related diseases and the benefits of the HPV vaccine to the parents or guardians and the school children is needed. Health education interventions incorporated with healthcare policy implementation to remove the barriers of vaccine acceptance may be useful in facilitating HPV acceptance and coverage.

## Figures and Tables

**Table 1 vaccines-14-00053-t001:** Baseline characteristics of participants (*n* = 943).

Factors	*n* (%)	Acceptance of HPV Vaccine for Their Children		*p*-Value
		No (*n* = 280) *n* (%)	Yes (*n* = 663) *n* (%)	
Gender				0.051
Male	225 (23.9)	81 (28.9)	144 (21.7)	
Female	715 (75.8)	198 (70.7)	517 (78)	
Other	3 (0.3)	1 (0.4)	2 (0.3)	
Age (year)				0.775
<30	128 (13.6)	44 (15.7)	84 (12.7)	
30–39	236 (25.0)	67 (23.9)	169 (25.5)	
40–49	424 (45.0)	122 (43.6)	302 (45.6)	
50–59	127 (13.5)	39 (13.9)	88 (13.3)	
>60	28 (3.0)	8 (2.9)	20 (3.0)	
Area				<0.001 *
1	210 (22.3)	36 (17.2)	174 (82.8)	
2	121 (12.8)	33 (27.3)	88 (72.7)	
3	523 (55.5)	198 (37.8)	325 (62.2)	
4	89 (9.4)	13 (14.6)	76 (85.4)	
Relationship with the children				1
Parent	815 (86.4)	242 (86.4)	573 (86.4)	
Non-parent	128 (13.6)	38 (13.6)	90 (13.6)	
Marital Status				0.396
Single	45 (6.0)	8 (4.8)	37 (6.4)	
Married	622 (83)	145 (86.8)	477 (82)	
Widowed	33 (4.4)	4 (2.4)	29 (5.0)	
Divorced	49 (6.5)	10 (6.0)	39 (6.7)	
Types of school				<0.001 *
Central governmental school	436 (46.2)	62 (22.1)	374 (56.4)	
Private school	242 (25.7)	84 (30.0)	158 (23.8)	
Local governmental school	265 (28.1)	134 (47.9)	131 (19.8)	
Religion				0.334
Buddhism	782 (82.9)	231 (82.5)	551 (83.1)	
Islam	152 (16.1)	44 (15.7)	108 (16.3)	
Christianity and others	9 (0.9)	5 (1.8)	4 (0.7)	
Occupation				0.051
Government officer	99 (10.5)	20 (7.1)	79 (11.9)	
Government employee	68 (7.2)	20 (7.1)	48 (7.3)	
Private company employee	146 (15.5)	48 (17.1)	98 (14.8)	
Freelance	320 (34)	87 (31.1)	233 (35.2)	
Business owner	291 (30.9)	96 (34.3)	195 (29.5)	
Others	18 (1.9)	9 (3.2)	9 (1.4)	
Education Level				0.018 *
Below Bachelor	464 (49.2)	156 (55.7)	308 (46.5)	
Bachelor	440 (46.7)	117 (41.8)	323 (48.7)	
Above Bachelor	39 (4.1)	7 (2.5)	32 (4.8)	
Income (THB)				0.639
<10,000	148 (15.7)	45 (16.1)	103 (15.5)	
10,000–25,000	412 (43.7)	125 (44.6)	287 (43.3)	
25,001–50,000	214 (22.7)	64 (22.9)	150 (22.6)	
>50,000	169 (17.8)	46 (16.4)	123 (918.5)	
Child’s healthcare cost entitlement				<0.001 *
Universal healthcare coverage	688 (73.2)	203 (72.5)	485 (73.5)	
Governmental reimbursable	102 (10.9)	16 (5.7)	86 (13)	
Health insurance of school	150 (16)	61 (21.8)	89 (13.5)	
Presence of healthcare personnel in family				0.002 *
No	859 (91.1)	268 (95.7)	591 (89.1)	
Yes	84 (8.9)	12 (4.3)	72 (10.9)	
Parents’ experiences receiving vaccines				<0.001 *
Yes	121 (12.9)	18 (6.4)	103 (15.6)	
No	820 (87.1)	262 (93.6)	558 (84.4)	
Knowledge level				0.001 *
Low	257 (27.5)	99 (35.7)	158 (24)	
Moderate	490 (52.4)	128 (46.2)	362 (54.9)	
High	189 (20.2)	50 (18.1)	139 (21.1)	
Accessibility level				<0.001 *
Low	498 (53)	195 (70.1)	303 (45.8)	
Moderate	148 (15.8)	28 (10.1)	120 (18.2)	
High	293 (31.2)	55 (19.8)	238 (36)	
Attitude				<0.001 *
Negative	16 (1.7)	13 (4.7)	3 (0.5)	
Neutral	874 (93.1)	251 (90.3)	623 (94.3)	
Positive	49 (5.2)	14 (5)	35 (5.3)	

* *p* < 0.05, Chi-square.

**Table 2 vaccines-14-00053-t002:** Multivariate logistic regression analysis of factors associated with parental adoption of HPV vaccine for their children.

Variables	Crude OR (95% CI)	Adj.OR (95% CI)	P (Wald’s Test)	P (LR-Test)
Type of school (Ref. = private school)				<0.001 *
Central governmental school	3.36 (2.31, 4.92)	3.10 (1.99, 4.86)	<0.001	
Local governmental school	0.56 (0.39, 0.80)	0.66 (0.43, 0.99)	0.047	
Education administration area (Ref. = 3)				0.05
1	2.94 (1.99, 4.44)	1.85 (1.16, 3.01)	0.011	
2	1.57 (1.02, 2.47)	0.96 (0.56, 1.66)	0.882	
4	3.57 (2.00, 6.89)	1.66 (0.84, 3.47)	0.158	
Presence of healthcare personnel in families	2.74 (1.52, 5.38)	2.50 (1.29, 5.22)	0.010	0.006 *
Child’s healthcare cost entitlement (Ref. = universal healthcare coverage)				0.027 *
Governmental reimbursement	2.22 (1.30, 4.01)	1.41 (0.78, 2.70)	0.272	
Health insurance of school	0.60 (0.41, 0.87)	0.61 (0.40, 0.94)	0.025	
Knowledge (Ref. = Low)				0.021 *
Moderate	1.79 (1.30, 2.47)	1.37 (0.95, 1.97)	<0.001	
High	1.73 (1.15, 2.61)	1.90 (1.20, 3.02)	<0.001	
Accessibility (Ref. = Low)				<0.001 *
Moderate	2.72 (1.76, 4.33)	2.58 (1.61, 4.24)	<0.001	
High	2.74 (1.95, 3.90)	2.25 (1.54, 3.32)	<0.001	
Attitudes (Ref. = Negative)	3.49 (2.19, 5.67)	3.13 (1.86, 5.34)	<0.001	0.005 *
Neutral	10.66 (3.40, 46.79)	7.69 (2.12, 37.84)	0.004	
Positive	10.83 (2.97, 52.80)	9.04 (2.11, 50.27)	0.005	

* *p* < 0.05, adjusted with all variables analyzed.

**Table 3 vaccines-14-00053-t003:** Thematic analysis.

Themes	Codes
1. There was a persistent misunderstanding that only females were the target population needing HPV vaccination. It was not necessary and had no benefit in males.	- It has been known that HPV vaccine is beneficial and recommended only to women, but it has no benefit to be recommended in males. - Because the HPV vaccine has been called the general Thai term of “cervical cancer prevention vaccine” and a men has no uterus, it will not be necessary for males at all. - Public health campaigns on HPV vaccination have focused markedly on cervical cancer prevention; much less information on vaccination for other HPV-related cancers or men. “At first, I understood that it was administered only to women, but later I found out that it could also be given to men to prevent HPV infection and related cancers for their sexual partners.” (P1) “It’s for prevention of cervical cancers given in pre-teens. I didn’t know it clearly because I didn’t have a daughter. So, I wasn’t interested.” (P2)
2. Vaccine safety was a major concern among the parents.	- HPV vaccine was considered a new vaccine by Thai people. - Giving the vaccine to young children caused major safety concerns among the parents. - There was a significant concern regarding the HPV vaccine’s safety like that of the COVID-19 vaccine. - Inadequate vaccine knowledge about the benefits of the vaccine over adverse effects led to concerns and hesitancy. - The safety of the HPV vaccine has not been strongly confirmed by healthcare professionals. “If I have my child vaccinated, will something happen?” and “Will it be dangerous?” (P7) “I’m also concerned about the adverse effects like those we have heard about COVID-19 vaccine. I, myself, think HPV vaccine is a newly-developed vaccine. I am worried about its side effects. Payment for vaccine cost is not an issue, but I want to know what the side effects the children may experience after being vaccinated.” (P2)
3.1 Unclear understanding and limited reliable public health information sources confirming the benefits of HPV vaccine. 3.2 Healthcare professionals were considered the most reliable sources of vaccine information and required for the people.	- Most of the available information concerning the benefit of HPV vaccine came only from online media, mostly unofficial, pamphlets, and health informing charts in hospitals. - Very limited discussion for clarification about the issues of concern about the vaccine with healthcare professionals. - People relied strongly on information and advice from healthcare professionals. - Community healthcare providers possibly provided health education on the HPV vaccine to reduce vaccine hesitancy and facilitate the acceptance of HPV vaccination. “I remembered that I had heard about it in short clips through online social medias. I didn’t pay much attention at that time but I was surprised to learn that there was a vaccine for prevention cervical cancer. Since I am a man, I am not interested in the vaccine but I’ve been known it more often over the past year.” (P10) “I often see medical students come to visit our village like village health volunteers, but they never discuss this topic. I think if they can promote this vaccine by explaining its benefit and adverse effects, it’s a more direct form of outreach to the target people.” (P11)
4. The appropriate timing of effective HPV vaccination.	- There were misunderstandings that The vaccine was most effective when the reproductive system of a female has fully developed. Many participants assumed the vaccine should be administered to female children when they started menstruation, because at that time, the uteruses were matured. - That the HPV vaccine should be given prior to the initiation of sexual relations was the unknown knowledge among the people. “I think it should be before menstruation, because after it starts, the uterus changes, right? That’s just my thought—I don’t have any scientific knowledge about the issue. But I think HPV vaccine may help protect the uterus, ovaries, and fallopian tubes. That’s why I think it should be given.” (P8) “In my opinion, secondary school children seem more appropriate than those in primary school which are too young. I think it’s better to wait until their sex hormones are fully developed before getting the vaccine.” (P6)
5. The experience with HPV vaccination of other people influenced the parents’ acceptance of the vaccine for their children.	- Other people’s positive experience of HPV vaccination influenced decision-making very much. - The absence of severe side effects reported after vaccination among other people led to greater confidence to have the children vaccinated. “The younger sibling was vaccinated first before the older one at their school. Since no side effect occurred in the younger one, we let the older one had it too. Only a little of a headache experienced by the older child afterward, but it resolved finally. I’m not so worried, but it seems to affect people differently. So, now both of our children have completed two doses of HPV vaccine.” (P9)
6. Limited reliable information available and the current campaigns on HPV vaccination were less strong.	- The vaccine campaign did not reach the target group directly and broadly. - Schools or community events to meet the parents, door to-door visits by community healthcare professionals during the community dwellers’ free time, and official social media platforms should be reinforced to facilitate vaccine acceptance. - Highly reliable information about the vaccine should be provided, best by the public healthcare providers and team, rather than unofficial social media available currently. - The community leaders were possibly key people for promoting vaccine acceptance also. “Local public health organization should prioritize this issues, because cervical cancer vaccination campaigns are rarely seen.” (P8) “We didn’t know the vaccine was beneficial to prevent cervical cancer. We also didn’t understand the details—like how many strains it could protect. I’d like the healthcare providers to organize knowledge-sharing sessions, so we can gain a clearer understanding.” (P7)
7. High cost of vaccine and limited government funding support limited vaccine accessibility.	- The cost of a full-course of the HPV vaccine was higher than affordable in a group population. - Inadequate financial support for vaccine cost from school health insurance for the vaccine cost. - Thai national universal healthcare cost coverage scheme can fund the vaccine cost for the people who fulfill the requirements endorsed by the national healthcare policy. It should be no more than 500–1000 THB per dose of vaccination which can be affordable by majority of people. Also, I need to know more about the potential side effects of the vaccine.” (P2) “I think vaccine cost should not be more than 1000 THB.” (P7) “The vaccine cost should be comparable to influenza vaccine, which was used to be 700–900 THB, but now it’s around 450–500.” (P5)
8. Geographical barriers limited vaccine access.	- Long distance from healthcare centers and difficulties in traffic communication precluded some people to receive HPV vaccination. School-based vaccine administration can be a better way for people to access the vaccine from a young age. “It’s difficult to receive the vaccine because a hospital or healthcare center is far away from home. The district hospital is over ten kilometers, while the provincial hospital in Hat Yai or Songkhla is over one hundred kilometers away from my home. If the vaccine isn’t distributed to the villages by hospital staff, it’s very difficult to access. Some people are inconvenient to travel such a long distance to receive the vaccine.” (P4)

## Data Availability

The study methods, analyses, and results are totally described in the published article.

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
