# Peer review of "Parents’ or Guardians’ Decisions on Human Papillomavirus Vaccine Acceptance for School Children in a Southern Province of Thailand: A Mixed-Method Study"

_vaccines, 2025, doi:10.3390/vaccines14010053_

Round 1
Reviewer 1 Report
Comments and Suggestions for Authors
The first three references cited are not source references for worldwide cervical cancer incidence. Reference four would be better.
The description of Thai cervical cancer statistics needs to be referenced:
"In Thailand, cervical cancer has been the second most common cancer among women following breast cancer. It affects mostly the women grouped in low to middle socio-economic class. Intensive cervical cancer screening in community as well as treatment in institution has been long conducted in Thailand to lessen the devastating situation."
HPV vaccine coverage rate should be discussed in the Introduction rather than the Discussion.
Reference to previous research supporting this study is missing in the introduction. A number of reasons for vaccine hesitancy are described in the introduction, but no references are included to support the importance of these factors.
The tables could be clearer by removing some of the lines - and separating subsections.
What does this study add to the literature over and about ref 17?
Comments on the Quality of English LanguageThe English and sentence structure is awkward throughout and bordered on being ambiguous. The paper requires the assistance of a native speaker.
Author Response
Response to Reviewer 1
Dear reviewer,
Thank you very much for comments. Please our responses below.
Regards,
------------------------------------------------------------------------------------------------------------
Reviewer 1
Comments and Suggestions for Authors
The first three references cited are not source references for worldwide cervical cancer incidence. Reference four would be better.
Response: We removed them and replaced them by an updated reference [https://www.who.int/news-room/fact-sheets/detail/cervical-cancer][Line33-34].
The description of Thai cervical cancer statistics needs to be referenced:
"In Thailand, cervical cancer has been the second most common cancer among women following breast cancer. It affects mostly the women grouped in low to middle socio-economic class. Intensive cervical cancer screening in community as well as treatment in institution has been long conducted in Thailand to lessen the devastating situation."
Response: The paragraph was revised to mention the most updated information and a reference was cited [Line 34-38].
HPV vaccine coverage rate should be discussed in the Introduction rather than the Discussion.
Response: This was addressed in introduction as advice [Line 57-63].
Reference to previous research supporting this study is missing in the introduction. A number of reasons for vaccine hesitancy are described in the introduction, but no references are included to support the importance of these factors.
Response: The papers discussed vaccine hesitancy were cited following your advice [ 9-18, Line 56].
The tables could be clearer by removing some of the lines - and separating subsections.
Response: Thank you for your suggestions. We followed your advice to revise the tables.
What does this study add to the literature over and about ref 17?
Response: The results of this study stressed on the correct knowledge regarding vaccine safety as the first priority, followed by its efficacy, formation positive attitude towards HPV vaccine, as well as the lowering barriers for vaccine access (vaccine cost and routes to access) were associated with high parents’ acceptance of the vaccine for their children. These results were generally reported in may literature including Ref 17. However, we further added that the parents’ principle concern about vaccine safety, especially for the young female children, and its efficacy or necessity for male. We think they will be the targets for planning education intervention to facilitate the vaccine acceptance and higher coverage in Thailand or the others.
Reviewer 2 Report
Comments and Suggestions for Authors
The manuscript tackles an important and timely public health issue in Thailand—parental acceptance of the HPV vaccine among school-aged children. The mixed-methods approach enhances the study by integrating both quantitative and qualitative perspectives, supported by a robust sample size. The topic is highly relevant and aligns well with national and global cervical cancer prevention efforts. Notably, the study benefits from its large and diverse participant group, including both parents and guardians. The identification of persistent misconceptions regarding HPV vaccination in males offers a particularly valuable practical insight. In addition, the manuscript effectively highlights the critical role of healthcare professionals and the importance of vaccine accessibility in shaping parental decision-making.
However, the manuscript requires substantial revisions to improve clarity, methodological transparency, data presentation, and English language quality. Several results and methodological procedures need clearer explanation, and the discussion would benefit from deeper integration of mixed-methods findings.
Comments:
- Abstract
- Several sentences need rephrasing for clarity; avoid unnecessary background details.
- Clarity and Structure
- The manuscript is lengthy and sometimes repetitive. Several sections, particularly the Introduction and Discussion, contain background information that could be streamlined.
- Some paragraphs mix results, interpretations, and literature, making the narrative difficult to follow.
- Please reorganise the sections to enhance coherence, remove any redundant background information, and ensure that the results and their interpretations are clearly distinguished.
- Please correct Human papilloma virus” to Human papillomavirus” throughout the manuscript.
- Use the full name on first mention and consistently use the abbreviation thereafter.
- Introduction: Multiple citations describe similar statistics; consider condensing.
- Methodology Needs Clearer Justification
- The sample size calculation section is overly detailed yet unclear in parts. It is difficult to follow the rationale for selecting multiple scenarios.
- Stratified and cluster sampling procedures require a clearer description (e.g., how schools were selected, how strata were defined).
- Please provide a concise, step-by-step explanation of sampling, recruitment, and the sequencing of the mixed methods.
- Questionnaire Development and Validation
- You state the IOC and Cronbach’s alpha values, but more information is needed about the development process.
- It is unclear how many items each subscale contained and how scores were calculated.
- Please include scoring methods, item counts per domain, and cut-off definitions.
- Results Presentation
- The overall structure of the results is clear, but the tables are difficult to interpret due to formatting issues and a lack of explanatory footnotes. Please improve table formatting and ensure consistency in percentages and P-values.
- The qualitative results are useful, but themes require clearer explanations with quotations or examples to enhance credibility.
- Add a brief quotation for each qualitative theme to enhance rigour.
- In Table 1, the total number is 943, not 944. Please confirm throughout the whole manuscript.
- In Table 1, it is easier to include the total number in the header for each column. For example, for the total participants “n = 943”, the willingness to receive the HPV vaccine is as follows: Yes, “n = 280”; No, “n = 663”.
- Tables:
- Ensure consistent decimal formatting.
- Clarify abbreviations on first use (CGS, LGS, PS, UHC).
- Interpretation of Logistic Regression
- Some adjusted odds ratios have wide confidence intervals or lose significance after adjustment; the discussion should highlight these nuances.
- Interpretation sometimes overstates causality.
- Please reframe results as associations, avoid causal language, and discuss potential confounding factors.
- Integration of Quantitative and Qualitative Findings
- Mixed-methods integration is minimal. Currently, the two strands appear as parallel but independent analyses. Please add a subsection discussing how qualitative findings help explain quantitative associations (e.g., misunderstandings about male vaccination, safety concerns, accessibility issues).
- Discussion: Some sections repeat results; others rely heavily on literature rather than study findings.
- Conclusion: Should be more concise and focused on the study’s implications rather than repeating background information.
English Language and Style
The manuscript contains frequent grammatical, syntactic, and word-choice issues, and several sentences are excessively long or unclear. A thorough professional English language edit is strongly recommended to improve clarity, coherence, and overall readability before the manuscript can be considered for publication.
Author Response
Response to Reviewer 2
Dear reviewer,
Thank you very much for comments. Please our responses below.
Regards,
------------------------------------------------------------------------------------------------------------
Reviewer 2
The manuscript tackles an important and timely public health issue in Thailand—parental acceptance of the HPV vaccine among school-aged children. The mixed-methods approach enhances the study by integrating both quantitative and qualitative perspectives, supported by a robust sample size. The topic is highly relevant and aligns well with national and global cervical cancer prevention efforts. Notably, the study benefits from its large and diverse participant group, including both parents and guardians. The identification of persistent misconceptions regarding HPV vaccination in males offers a particularly valuable practical insight. In addition, the manuscript effectively highlights the critical role of healthcare professionals and the importance of vaccine accessibility in shaping parental decision-making.
However, the manuscript requires substantial revisions to improve clarity, methodological transparency, data presentation, and English language quality. Several results and methodological procedures need clearer explanation, and the discussion would benefit from deeper integration of mixed-methods findings.
Comments:
Abstract
Several sentences need rephrasing for clarity; avoid unnecessary background details.
Response: The abstract was revised for clarity and conciseness as advice.
Clarity and Structure
The manuscript is lengthy and sometimes repetitive. Several sections, particularly the Introduction and Discussion, contain background information that could be streamlined.
Some paragraphs mix results, interpretations, and literature, making the narrative difficult to follow.
Please reorganise the sections to enhance coherence, remove any redundant background information, and ensure that the results and their interpretations are clearly distinguished.
Please correct Human papilloma virus” to Human papillomavirus” throughout the manuscript.
Use the full name on first mention and consistently use the abbreviation thereafter.
Response: Thank you for your notice and advice. We revised the manuscript throughout.
Human papillomavirus replaced all the former erroneous writing.
Full name used at first mentioned, and abbreviations consistently used afterwards
Introduction: Multiple citations describe similar statistics; consider condensing.
Response: They were revised and condensed.
Methodology Needs Clearer Justification
The sample size calculation section is overly detailed yet unclear in parts. It is difficult to follow the rationale for selecting multiple scenarios.
Response: The method of sample size calculation was specified [See 2.3 sample size calculation]
Stratified and cluster sampling procedures require a clearer description (e.g., how schools were selected, how strata were defined).
Response: Stratified and cluster sampling method was described in steps [See 2.4 Sampling technique, Line 97-109]
Please provide a concise, step-by-step explanation of sampling, recruitment, and the sequencing of the mixed methods.
Response: The sequence of the procedures was described in data collection subsection [See 2.4 Sampling technique, Line 97-109 and 2.6 Data collection, Line 139-148].
Questionnaire Development and Validation
You state the IOC and Cronbach’s alpha values, but more information is needed about the development process.
It is unclear how many items each subscale contained and how scores were calculated.
Please include scoring methods, item counts per domain, and cut-off definitions.
Response: The scale and scoring method of individual instrument were provided [Please see subsection Questionnaire development and validation, Line 116-131 and supplementary tables foot notes].
Results Presentation
The overall structure of the results is clear, but the tables are difficult to interpret due to formatting issues and a lack of explanatory footnotes. Please improve table formatting and ensure consistency in percentages and P-values.
Response: All table formats were revised to present much clearer understandable
The qualitative results are useful, but themes require clearer explanations with quotations or examples to enhance credibility.
Response: The themes and codes were revised (yellow highlights), and examples of quotation were added (Blue highlights) [See results (Table 3 thematic analysis). The rest of quotations were shown in supplementary section B].
Add a brief quotation for each qualitative theme to enhance rigour.
Response: A brief quotation was added to each theme.
In Table 1, the total number is 943, not 944. Please confirm throughout the whole manuscript.
In Table 1, it is easier to include the total number in the header for each column. For example, for the total participants “n = 943”, the willingness to receive the HPV vaccine is as follows: Yes, “n = 280”; No, “n = 663”.
Response: Sorry for the errors. The number was revised. The n=280 and n=663 were added to the table headers of each column according to your advice.
Tables:
Ensure consistent decimal formatting.
Clarify abbreviations on first use (CGS, LGS, PS, UHC).
Response: Thank you for your notice. We rechecked and revised their presentations [See 2.4 Sampling techniques-the full names were used at first presentations, and the followings in abbreviations].
Interpretation of Logistic Regression
Some adjusted odds ratios have wide confidence intervals or lose significance after adjustment; the discussion should highlight these nuances.
Interpretation sometimes overstates causality.
Please reframe results as associations, avoid causal language, and discuss potential confounding factors.
Response: Thank you your meaningful advice. We revised the discussion throughout following your suggestions.
Integration of Quantitative and Qualitative Findings
Mixed-methods integration is minimal. Currently, the two strands appear as parallel but independent analyses. Please add a subsection discussing how qualitative findings help explain quantitative associations (e.g., misunderstandings about male vaccination, safety concerns, accessibility issues).
Discussion: Some sections repeat results; others rely heavily on literature rather than study findings.
Conclusion: Should be more concise and focused on the study’s implications rather than repeating background information.
Response: Thank you for your suggestions. We revised the discussion and conclusion. Also, a subsection discussing the integrated associations of the two study methods was added [Line 305-318].
Reviewer 3 Report
Comments and Suggestions for Authors
Dear authors,
Your manuscript addresses an important topic and is supported by current references. The research questions are clearly formulated. There are few points that could be improved for readers:
Introduction
The introduction is clear and concise. It describes the knowledge gap and rationale for the study. References are relevant and up to date.
Materials and Methods
This section presents important details, but as suggestions: there is not any international guideline checklist followed to strengthen the manuscript. Please, consider including one. Additionally,
Results
The results section is comprehensive but may benefit from improved readability: The tables have many subdivisions lines. Please, consider revising this.
Discussion
The discussion provides insightful interpretation and situates findings within the broader literature. However, it is very truncated, with some paragraphs that are big and with minor grammar errors. Please, make a detailed revision of that.
Comments on the Quality of English LanguageThe manuscript contains several errors related to punctuation and grammar. It is important to carefully review the entire document to correct these issues.
Author Response
Response to Reviewer 3
Dear reviewer,
Thank you very much for comments. Please our responses below.
Regards,
------------------------------------------------------------------------------------------------------------
Dear authors,
Your manuscript addresses an important topic and is supported by current references. The research questions are clearly formulated. There are few points that could be improved for readers:
Introduction
The introduction is clear and concise. It describes the knowledge gap and rationale for the study. References are relevant and up to date.
Response: Thank you for your comment.
Materials and Methods
This section presents important details, but as suggestions: there is not any international guideline checklist followed to strengthen the manuscript. Please, consider including one. Additionally,
Response: We added a standard check list (STROBE) as your advice.
Results
The results section is comprehensive but may benefit from improved readability: The tables have many subdivisions lines. Please, consider revising this.
Response: We revised the table format for more readability.
Discussion
The discussion provides insightful interpretation and situates findings within the broader literature. However, it is very truncated, with some paragraphs that are big and with minor grammar errors. Please, make a detailed revision of that
Response: Thank you for your notice. We also realized the points of grammatical errors you raised. We thoroughly revised the discussion section for clarity and readability.
Round 2
Reviewer 2 Report
Comments and Suggestions for Authors
The authors have formally acknowledged all major categories of comments and have made noticeable efforts to revise the manuscript.
However, there are comments regarding the table design:
- Tables are dense and difficult to read
- Ensure there is proper spacing between items in the tables for better readability.
- All tables should be self-explanatory and include clear footnotes.
Author Response
Dear reviewer,
We revised the tables and their associated footnotes according to your advice.
Thank you very much again for your full support and advice.
Regards,